# Fenpicoxamid-Imprinted Surface Plasmon Resonance (SPR) Sensor Based on Sulfur-Doped Graphitic Carbon Nitride and Its Application to Rice Samples

**DOI:** 10.3390/mi15010006

**Published:** 2023-12-19

**Authors:** Şule Yıldırım Akıcı, Bahar Bankoğlu Yola, Betül Karslıoğlu, İlknur Polat, Necip Atar, Mehmet Lütfi Yola

**Affiliations:** 1Department of Nutrition and Dietetics, Faculty of Health Sciences, Hasan Kalyoncu University, Gaziantep 27000, Turkey; sule.yakici@std.hku.edu.tr (Ş.Y.A.); ilknur.polat@hku.edu.tr (İ.P.); 2Department of Engineering Basic Sciences, Faculty of Engineering and Natural Sciences, Gaziantep Islam Science and Technology University, Gaziantep 27000, Turkey; bahar.bankogluyola@gibtu.edu.tr; 3Department of Gastronomy and Culinary Arts, Faculty of Tourism, Hasan Kalyoncu University, Gaziantep 27000, Turkey; betul.kokay@hku.edu.tr; 4Department of Chemical Engineering, Faculty of Engineering, Pamukkale University, Denizli 20160, Turkey; natar@pau.edu.tr

**Keywords:** fenpicoxamid, surface plasmon resonance, molecularly imprinting, food analysis

## Abstract

This research attempt involved the development and utilization of a newly designed surface plasmon resonance (SPR) sensor which incorporated sulfur-doped graphitic carbon nitride (S-g-C_3_N_4_) as the molecular imprinting material. The primary objective was to employ this sensor for the quantitative analysis of Fenpicoxamid (FEN) in rice samples. The synthesis of S-g-C_3_N_4_ with excellent purity was achieved using the thermal poly-condensation approach, which adheres to the principles of green chemistry. Afterwards, UV polymerization was utilized to fabricate a surface plasmon resonance (SPR) chip imprinted with FEN, employing S-g-C_3_N_4_ as the substrate material. This process involved the inclusion of N,N′-azobisisobutyronitrile (AIBN) as the initiator, ethylene glycol dimethacrylate (EGDMA) as the cross-linker, methacryloylamidoglutamic acid (MAGA) as the monomer, and FEN as the analyte. After successful structural analysis investigations on a surface plasmon resonance (SPR) chip utilizing S-g-C_3_N_4_, which was imprinted with FEN, a comprehensive investigation was conducted using spectroscopic, microscopic, and electrochemical techniques. Subsequently, the kinetic analysis applications, namely the determination of the limit of quantification (LOQ) and the limit of detection (LOD), were carried out. For analytical results, the linearity of the FEN-imprinted SPR chip based on S-g-C_3_N_4_ was determined as 1.0–10.0 ng L^−1^ FEN, and LOQ and LOD values were obtained as 1.0 ng L^−1^ and 0.30 ng L^−1^, respectively. Finally, the prepared SPR sensor’s high selectivity, repeatability, reproducibility, and stability will ensure safe food consumption worldwide.

## 1. Introduction

The World Health Organization foresees that the population will reach 9 billion in 2050. Although the world’s food demand has increased, agricultural areas have remained the same. This situation has made it a necessity to use these areas efficiently and improve the productivity and quality of agricultural products [1]. Especially, sustainability in agriculture is an important issue that must be discussed for our future. All these explain why pesticides have become an important starting point in increasing crop productivity, but their harms need to be investigated [2]. Pesticides generally include insecticides, fungicides, herbicides, and nematicides, all of which have found a place in agricultural practice [3]. Among them, fungicides can generally control fungal disease. However, many fungi have developed resistance to fungicides [4]. Pyrenophora tritici-repentis is a fungal phytopathogen responsible for the global occurrence of tan spot disease in wheat. In recent years, the development of resistance to this disease has been reported. Therefore, there was a need to develop a new fungicide for the management of this pathogen, and FEN is a new fungicide to be used in this field [5]. 

FEN, a significant propesticide, possesses minimal risk to both human health and the ecosystem. This compound is derived from the culture fluid of *Streptomyces* spp. and is utilized for the effective management of several plant fungal infections. In addition, it is a derivative of an antibiotic that has strong antifungal properties against a diverse array of fungal diseases. Additionally, it can be effectively employed to combat several grain pathogens. However, pesticide residues in food can harm human health, so attention should be paid to dosages for food safety, especially for newly discovered pesticides [6]. Maximum residue levels are legally permitted in food or animal feed, based on the lowest exposure to protect health. The European Union has made some regulations regarding the presence of FEN in foods. The acceptable daily intake (ADI), acute reference dose (ARfD), and acceptable operator exposure level of FEN are determined to be 0.05 mg/kg live weight/day, 1.8 mg/kg live weight, and 0.05 mg/kg live weight/day [7], respectively. Therefore, the identification of foodborne enteric pathogens holds significant significance in ensuring the safety of food intake in contemporary times.

Surface plasmons having two forms, including localized surface plasmons and polaritons, mean the coherent oscillations of conduction electrons on a silver (Ag) or gold (Au) surface under the quantized energies. These are significant for plasmonic sensor applications [7]. When an electromagnetic wave is applied to a plasmonic nanomaterial, evident electron oscillations occur on the plasmonic nanomaterial surface, and light energy is absorbed by the plasmonic nanomaterial surface. The surface plasmon polaritons propagate the oscillations of surface electrons on a silver (Ag) or gold (Au) surface. When p-polarized light enters into the metal surface at a greater angle, the total reflection case emerges, and the evanescent wave occurs on a silver (Ag) or gold (Au) surface [8,9]. Hence, free electrons on a silver (Ag) or gold (Au) surface can be excited to create surface plasmon waves. The detection mechanism employed in the SPR system relies on the identification of a shift in the resonance angle, which can be attributed to alterations in the refractive index [10]. SPR sensors have a short response time and can be easily integrated with microfluidics to develop a total analysis system called “lab-on-a-chip” that simultaneously features sample preparation, chemical analysis, and data evaluation [11]. In addition, it is possible to analyze binding and dissociation events simultaneously in SPR sensors [12,13]. Furthermore, the sensor system has a broad spectrum of applications in the field of food analysis, encompassing areas such as the detection of toxins and nutritional supplements [14].

The utilization of graphitic carbon nitride (g-C_3_N_4_) as a nanomaterial has gained prominence due to its notable characteristics, including exceptional chemical stability and non-toxic nature. G-C_3_N_4_ is classified as a metal-free semiconductor and, in contrast to graphene, exhibits a very porous morphology. The composition mostly comprises carbon (C) and nitrogen (N) atoms, with a small amount of hydrogen (H) atoms [15,16]. Thus, g-C_3_N_4_ is used in many fields, including photocatalysis, photovoltaics, and sensors [17,18]. The optical and electrical characteristics of g-C_3_N_4_ exhibit notable variations depending on the choice of precursors employed during its synthesis. Furthermore, it was observed that the utilization of the heteroatom in the synthesis of sulfur-doped graphitic carbon nitride, including the thermal poly-condensation method with a higher degree of polymerization, resulted in a material that exhibited enhanced environmental sustainability and a low cost [19]. 

The molecular imprinting technique aims to create selective materials with chemical functions by covalent or non-covalent interactions of functional monomers around the template molecule and subsequent polymerization. The molecular imprinting process includes a series of three distinct stages: (i) pre-complexation, (ii) polymerization, and (iii) template molecule removal. Consequently, the formation of polymeric cavities that are unique to the analyte molecule occurs. Molecularly imprinted polymers (MIPs) can be utilized in significant applications, such as separation and sensor [20]. MIPs are tough and resistant to high pressure and temperature, with higher physical strength compared to some biological structures such as proteins. They are also inert towards various chemicals (organic solvents and metal ions) [21,22].

The maximum residue limits (MRLs) of FEN in food products can pose a risk to human health. For this reason, ensuring food safety is more important, especially for newly marketed compounds, and sensitive and reliable analytical methods are urgently needed. This study presents a molecularly imprinted SPR sensor utilizing S-g-C_3_N_4_ for the detection and recognition of FEN in rice samples. Following the synthesizing process of S-g-C_3_N_4_ of exceptional purity, a molecularly imprinted SPR sensor was constructed and subsequently employed for the selective quantification of FEN levels in rice samples, yielding a high recovery rate. One of the important aspects of this study is that the developed sensor has a very sensitive measurement range, and is environmentally friendly, economical, and easily applicable. We also hope that this sensor will contribute to both food safety and environmental health by detecting the residue amounts of important pesticides such as FEN.

## 2. Materials and Methods

### 2.1. Materials

FEN, antibiotic UK 2A (UK-2A), antimycin A3 (AA), melamine (MEL), 2-hydroxyethylmethacrylate (HEMA), thiourea (THI), ethyl alcohol (EtOH), acetonitrile (ACN), and sodium chloride (NaCl) were procured from Sigma-Aldrich (St. Louis, MO, USA). The dilution solution utilized in this study was phosphate-buffered saline (PBS), with a concentration of 0.1 M and a pH of 6.0.

### 2.2. Instrumentation

The instrumental devices for morphological analyzes, such as JEOL 2100 TEM (Tokyo, Japan) for transmission electron microscopy (TEM), Rikagu Miniflex, X-ray diffractometer (Tokyo, Japan) for X-ray diffraction analysis (XRD), Bruker-Tensor 27 FTIR spectrometer (Tokyo, Japan) for Fourier-transform infrared spectroscopy (FTIR), PHI 5000 Versa Probe type X-ray photoelectron spectrometer (Tokyo, Japan/New York, NY, USA) for X-ray photoelectron spectroscopy (XPS), AFM Park NX10 (Tokyo, Japan) for atomic force microscopy (AFM), and Thermo Fisher Scientific UV-Vis/Vis Instrumentation for UV-Vis, were used with great care. The electrochemical impedance spectroscopy (EIS) and cyclic voltammetry (CV) measurements were conducted using the Gamry Reference 600 workstation from the Warminster, PA, USA. For kinetic analysis, the GenOptics SPR system from Calgary, AB, Canada was employed.

### 2.3. Preparation of g-C_3_N_4_ and S-g-C_3_N_4_ Nanomaterials

The synthesis of bulk g-C_3_N_4_ was completed by the thermal poly-condensation method [23]. The heating process at 600 °C was applied to MEL (5.0 g) in an alumina crucible with a rising rate of 10 °C min^−1^ during 30 min, and bulk g-C_3_N_4_ was stored during the cooling process. Then, it was ground into powder for analytical procedures. S-g-C_3_N_4_ was obtained in the presence of THI (10.0 g) by repeating the experimental procedures described above for g-C_3_N_4_.

### 2.4. SPR Chip Modification with S-g-C_3_N_4_ and the Development of FEN-Imprinted SPR Sensor Based on S-g-C_3_N_4_

Before initiating kinetic analyses, the surfaces of SPR chips underwent a cleaning step using acidic piranha solution (10.0 mL, 3:1 H_2_SO_4_:H_2_O_2_, *v*/*v*). Following immersion in the acidic piranha solution, the cleaning process continued in a shaking bath system for 20 min. Following this, the SPR chips underwent a drying procedure under nitrogen gas, rendering them suitable for subsequent modifications. During modification, a solution containing S-g-C_3_N_4_ nanomaterial (20.0 mg mL^−1^) was applied to the cleaned SPR chips and left at 25 °C for 15 min. This allowed for the modification process to occur through the strong affinity and interaction between gold and sulfur (S-g-C_3_N_4_/SPR) [22].

For the improvement in the FEN-imprinted SPR sensor based on S-g-C_3_N_4_, two separate polymerization solutions were prepared. Firstly, FEN-MAGA complex was prepared in PBS (0.1 M, pH 6.0) at a mole ratio of (2:1). On the other hand, a solution consisting of AIBN (20.0 mg), HEMA (2.0 mL), and EGDMA (4.0 mL) was made in a separate tube, with the addition of PBS (0.1 M, pH 6.0). 

The complex solution (1.0 mL) was gradually Introduced into the AIBN-HEMA-EGDMA solution (2.0 mL) over a period of 30 min. This final homogeneous solution was dropped onto the S-g-C_3_N_4_/SPR chip surface using the spin-coating method and then left at room temperature for 30 min. Following a 10 min UV polymerization period, an FEN-imprinted SPR sensor incorporating S-g-C_3_N_4_ was prepared (MIP/S-g-C_3_N_4_/SPR). NIP/S-g-C_3_N_4_/SPR was fabricated to showcase the imprinting selectivity by applying the same procedure above without the FEN molecule.

### 2.5. Sample Preparation, FEN Removal from MIP/S-g-C_3_N_4_/SPR, and Analysis Procedure

Rice samples (0.20 g) taken from a local food store were first crushed to powder and then transferred to the EtOH:ACN mixture (20.00 mL, 1:1, *v*/*v*) and mixed for approximately 30 min. After the centrifugation process (20 min at 10,000 rpm), the transparent part was transferred to the SPR cell for analysis.

After the prepared MIP/S-g-C_3_N_4_/SPR chip was attached to the SPR cell, a suitable desorption solution (0.1 M NaCl) was selected according to the interactions (electrostatic/hydrogen bond) between FEN and MAGA. For this aim, the MIP/S-g-C_3_N_4_/SPR chip was kept in a shaking bath system containing 0.1 M NaCl (10.0 mL) for 20 min of desorption time. After 20 min, the FEN-removed SPR chip was dried under nitrogen atmosphere, and kinetic analysis was carried out. The same treatment was replicated for the NIP/S-g-C_3_N_4_/SPR chip.

After the SPR chips, including MIP and NIP, were prepared, 0.1 M PBS (pH 6.0) solution was passed over the chip surface for 10 min for kinetic analysis (2.0 mL min^−1^ flow-rate). Afterwards, FEN solutions at increasing concentrations were interacted with the chip surface to reach a constant plateau region for 50 min (2.0 mL min^−1^ flow-rate). Finally, the “adsorption–desorption-regeneration” cycle was completed by desorption with 0.1 M NaCl. 

## 3. Results and Discussion

### 3.1. Characterizations of g-C_3_N_4_ and S-g-C_3_N_4_ Nanomaterials

XRD patterns were obtained for the phase structure investigations (Appendix A). According to the XRD patterns, XRD peaks that well described g-C_3_N_4_ emerged [24]. Two evident XRD peaks with no difference were observed for S-g-C_3_N_4_ and g-C_3_N_4_. The XRD peak at 27.38° was related to aromatic systems’ interlayer stacking, which corresponded to the (002) peak, attributing to the interlayer distance of d = 0.319 nm [25,26]. The XRD peak intensity at 27.38° for g-C_3_N_4_ was stronger than that of S-g-C_3_N_4_, providing the better crystallinity of g-C_3_N_4_. This situation was owing to MEL’s formation as an intermediate product with THI’s heating process [27]. The XRD peak having a small intensity at 13.22° was related to aromatic systems’ in-plane structural packing, which corresponded to the (100) peak, attributing to the interlayer distance of d = 0.681 nm [28]. According to these XRD patterns, these S-g-C_3_N_4_ and g-C_3_N_4_ materials produced with different precursors have the same crystal structure [29]. The fact that the S-g-C_3_N_4_ material has a lower intensity XRD peak was due to the doping of sulfur into the structure.

The electronic band structures of S-g-C_3_N_4_ and g-C_3_N_4_ nanomaterials were obtained (Appendix A). According to the UV-Vis spectra of S-g-C_3_N_4_ and g-C_3_N_4_, we saw that S-g-C_3_N_4_ had a higher absorption band in comparison with g-C_3_N_4_. In addition, the absorption band of g-C_3_N_4_ was slightly red-shifted [30,31]. The red shift and high absorption ability confirmed the improvement in the photoactivity of S-g-C_3_N_4_. 

The morphological studies of S-g-C_3_N_4_ and g-C_3_N_4_ were carried out by TEM images (Figure 1). According to these TEM images, layered structures with irregular pores were observed on S-g-C_3_N_4_ and g-C_3_N_4_. In fact, the number of irregular pores in the S-g-C_3_N_4_ material was higher than that in the g-C_3_N_4_ material [32]. In addition, both S-g-C_3_N_4_ and g-C_3_N_4_ had many nanosheet-sized layers and nanoparticles with different shapes [33]. Furthermore, the particles of g-C_3_N_4_ (Figure 1B) were thicker than those of S-g-C_3_N_4_ (Figure 1A) because of g-C_3_N_4′_s higher degree of polymerization [26].

The elemental constitutions of S-g-C_3_N_4_ and g-C_3_N_4_ nanomaterials were investigated using XPS measurements (Appendix A). Two XPS peaks at 284.53 and 288.08 eV on the C1s spectrum were attributed to carbon impurities and N=C–N, including sp^2^-bonding, respectively [34,35]. The XPS peaks at 399.06 eV on the N1s spectrum corresponded to the aromatic C–N=C, including sp^2^-bonding. The XPS peaks at 400.08 and 401.13 eV on the N1s spectrum were attributed to nitrogen bonded to carbon atoms in N–C_3_ and the C–N–H with the amino group, respectively. Finally, the last XPS peak at 404.28 eV on the N1s spectrum was related to π excitations [36]. On the O1s spectra, the XPS peak at 532.19 eV was related to the adsorbed CO_2_ and H_2_O on g-C_3_N_4,_ while the XPS peak on S-g-C_3_N_4_ was lower than that of g-C_3_N_4_, providing the presence of SO_4_^2−^ in S-g-C_3_N_4_ nanomaterial [15]. XPS peaks at 163.98, 165.39, and 170.08 eV on the S2p spectrum of S-g-C_3_N_4_ were corresponded to C–S bonds, N–S bonds, and the presence of SO_4_^2−^, respectively [19].

FTIR measurements were performed for the investigations of the chemical structures of S-g-C_3_N_4_ and g-C_3_N_4_ (Appendix A). According to Appendix A, there was no obvious difference between the FTIR spectra of the nanomaterials. The absorption bands at 1247, 1331, 1411, 1461, 1569, and 1641 cm^−1^ for S-g-C_3_N_4_ and g-C_3_N_4_ were attributed to the typical stretching of –C_6_N_7_ units [37]. The absorption bands at 805 and 891 cm^−1^ were corresponded to the specific triazine units, confirming CN heterocycles [38] and N–H deformation [39], respectively. A weak absorption band at 2382 cm^−1^ was related to the presence of CO_2_. The broad absorption peak at 2850–3550 cm^−1^ was corresponded to the presence of the vibrations of the H_2_O molecules and N–H groups. 

### 3.2. Electrochemical Characterizations of g-C_3_N_4_ and S-g-C_3_N_4_ Nanomaterials Modified Electrodes

To use the g-C_3_N_4_ and S-g-C_3_N_4_ nanomaterials as sensor materials, CV and EIS characterizations were performed to see the electrical conductivity performance of the synthesized nanomaterials. Electrochemical peaks, including anodic and cathodic signals, were observed (curve a of Figure 2A) using bare GCE. When the g-C_3_N_4_ modified glassy carbon electrode (g-C_3_N_4_/GCE) was used in the presence 5.0 mM [Fe(CN)_6_]^3−/4−^, more improved electrochemical peaks were observed (curve b of Figure 2A) owing to the g-C_3_N_4′_ graphite structure, with covalent bonds improving the electrical conductivity [40]. An evident increase in electrochemical peaks was observed on S-g-C_3_N_4_/GCE (curve c of Figure 2A) owing to sulfur incorporation into g-C_3_N_4_ nanomaterial, providing a more porous structure, causing electrochemical active defects [26]. Moreover, the i_p_ = 2.69 × 10^5^ A n^3/2^ D^1/2^ C v^1/2^ equation was applied for the calculation of electrode surface areas in the presence of 1.0 mM [Fe(CN)_6_]^3−^ and 0.073 ± 0.001 cm^2^, and 0.487 ± 0.004 cm^2^ and 0.738 ± 0.005 cm^2^ were provided for bare GCE, g-C_3_N_4_/GCE, and S-g-C_3_N_4_/GCE, respectively. Hence, the increase in electrochemical active defects has caused a significant increase in surface areas.

The charge transfer resistance (R_ct_) was investigated using EIS measurements (Figure 2B), and 33 ohm for bare GCE (curve a), 27 ohm for g-C_3_N_4_/GCE (curve b), and 21 ohm for S-g-C_3_N_4_/GCE (curve c) were obtained as the R_ct_ values, providing the highest electrochemical conductivity on S-g-C_3_N_4_/GCE in harmony with CV measurements. 

### 3.3. FTIR and AFM Studies of FEN-Imprinted Film on S-g-C_3_N_4_/SPR

The FTIR spectra of the FEN-imprinted SPR chip, including HEMA and MAGA, were recorded (Appendix A). FTIR peaks at 3598 cm^−1^ attributing to HEMA and MAGA, 3003 cm^−1^ corresponding to the –CH stretching of MAGA, 1695 cm^−1^ relating to the stretching of carboxyl–carbonyl, and 1444 cm^−1^ attributing to –COO– stretching were seen.

The surface thicknesses of bare SPR chip (Appendix A) and FEN-imprinted SPR chip based on S-g-C_3_N_4,_ including HEMA and MAGA (Appendix A), were recorded as 3.09 ± 0.06 and 24.73 ± 0.07 nm, respectively. These FTIR and AFM measurements confirmed the successful formation of FEN-imprinted film on S-g-C_3_N_4_/SPR chip.

### 3.4. Comparison with MIP/g-C_3_N_4_/SPR and MIP/S-g-C_3_N_4_/SPR Chips and pH Effect on SPR Signals in FEN Detection

To verify the electrochemical characterization results, the performance analyzes of MIP/g-C_3_N_4_/SPR and MIP/S-g-C_3_N_4_/SPR chips in the presence of 10.0 ng L^−1^ FEN were compared. The higher SPR signals on MIP/S-g-C_3_N_4_/SPR chip were obtained against 10.0 ng L^−1^ FEN in comparison with MIP/g-C_3_N_4_/SPR chip (Figure 3A). Thus, MIP/S-g-C_3_N_4_/SPR chip was used in subsequent analytical applications. 

pH is the crucial factor affecting SPR signal stability in optical sensor applications. In this study, MAGA monomer has two pKa values (pKa1: 2.10 and pKa2: 4.07) due to its chemical structure, and the carboxylic acid groups of MAGA transformed into anionic form at high pH values. Hence, MAGA-FEN affinity and interaction occurred to a high degree up to pH 6.0. On the contrary, this interaction decreased rapidly after pH 7.0, and significant decreases in SPR signals were observed (Figure 3B,C). Finally, the value of pH 6.0 was selected as the optimum medium pH in subsequent experiments.

### 3.5. Sensitivity of FEN-Imprinted SPR Chip Based on S-g-C_3_N_4_ (MIP/S-g-C_3_N_4_/SPR)

After preparing the FEN-imprinted SPR chip based on S-g-C_3_N_4_ (MIP/S-g-C_3_N_4_/SPR), the standard calibration graph was first generated. The purpose of recording the standard calibration graph was to test the operability of the sensor system and to find the linear range in which the SPR chip responded to the template FEN molecule. First, 0.1 M PBS (pH 6.0) solution was given to the SPR cell, where MIP/S-g-C_3_N_4_/SPR was located for 10 min, and then increasing concentrations of FEN standard solutions were interacted with the MIP/S-g-C_3_N_4_/SPR chip surface for 50 min. After reaching a plateau region, 0.1 M PBS (pH 6.0) solution was again interacted with the MIP/S-g-C_3_N_4_/SPR chip surface, thus the adsorption–desorption-regeneration cycle for each standard FEN solution was completed (Figure 4). The calibration equation of y (ΔR) = 1.0345x (C_FEN_, ng L^−1^) − 0.4419 was obtained when the obtained ΔR values were plotted against the standard FEN solutions. Finally, LOQ and LOD values were determined as 1.0 ng L^−1^ and 0.30 ng L^−1^, respectively (see Appendix A for the equations).

Classical chromatographic methods have been developed for the analysis of fungicides in the literature, and the selective measurements can be achieved by separating fungicides at different times in these techniques [41,42,43]. These methods are time-consuming and use excessive consumables and always require professional personnel. In this study, a low-cost measurement SPR system was developed based on these problems encountered in fungicide determinations. The SPR system is generally an affinity-based measurement system. In this SPR sensor system, molecularly imprinted polymers as recognition agents were used because of a low-cost and selective system. In addition, since the synthesis of sulfur-doped graphitic carbon nitride was successfully carried out by the thermal poly-condensation method, it can be said that the developed sensor was a suitable method for Green Chemistry thanks to zero waste generation. The developed SPR sensor system was also economical, easily prepared, and a more sensitive SPR sensor system based on S-g-C_3_N_4_ and MIP with an LOD of 0.30 ng L^−1^ was presented to the literature world when compared to other techniques. As a result, thanks to this developed SPR sensor, the propesticide analysis can be performed quickly from food samples, and some diseases caused by pesticide exposure can be diagnosed earlier.

### 3.6. Recovery

We expect a developed analytical method to provide analysis results with high accuracy and selectivity when it is used for analysis from real samples. To verify this accuracy and selectivity, recovery values are calculated by adding standard solutions at a certain concentration to real samples. In this study, the prepared rice samples for analysis were divided into four equal parts. Then, FEN standard solutions (2.00, 4.00, and 6.00 ng L^−1^) were added in increasing concentrations to all parts except the first part. 

Finally, these four equal solutions were mixed with 0.1 M PBS at a pH of 6.0, resulting in equal volumes. Following the analysis of four rice samples containing FEN using the constructed MIP/S-g-C_3_N_4_/SPR sensor, the recovery values were then determined and shown in Appendix A. After these four rice samples containing FEN were analyzed by the developed MIP/S-g-C_3_N_4_/SPR sensor, the recovery values were calculated (Appendix A). Based on the data presented in Appendix A, values close to 100% demonstrated that an SPR sensor with high selectivity was successfully prepared.

### 3.7. Selectivity, Stability, Reproducibility, and Repeatability of MIP/S-g-C_3_N_4_/SPR

During the FEN determination of the MIP/S-g-C_3_N_4_/SPR sensor, selectivity tests were carried out for confirming its high specificity in the presence of agents (UK-2A and AA). Thus, the obtained ΔR signals for MIP-based SPR chip (Figure 5A) and the NIP-based SPR chip (Figure 5B) were recorded. The values of the selectivity coefficient (k) and relative selectivity coefficient (k’) were computed using these ΔR signals (Appendix A). The specific nano-cavities belonging to FEN in the polymeric network caused more selectivity for FEN analysis in comparison with NIP SPR chip. In addition, the prepared SPR sensor based on MIP can be used more selectively and effectively in the presence of other agents with possible matrix effects.

SPR signal measurements of a prepared MIP-based SPR chip for a long-term stability study in the presence of 10.0 ng L^−1^ FEN for 10 weeks were examined. The ΔR values at the end of the tenth week were almost exactly the same as the ΔR values at the end of the first week. This showed that the MIP-based SPR sensor that was made is very stable.

For the reproducibility study, 15 different FEN-imprinted SPR sensor chips were prepared in the presence of MAGA, AIBN, and EGDMA, and the relative standard deviation (RSD) values of the obtained ΔR values by interacting each SPR chip with 10.0 ng L^−1^ FEN solution were determined as 0.94%, verifying the high reproducibility of the preparation process of FEN-imprinted SPR sensor chips.

Among the critical analytical performances in SPR sensor applications, repeatability stands out as paramount. For this aim, six consecutive cycles, including adsorption–desorption-regeneration, were finished by the MIP/S-g-C_3_N_4_/SPR sensor in the presence of 10.0 ng L^−1^ FEN. The obtained ΔR values at the end of each cycle were recorded, and the RSD values of 0.37% confirmed the high repeatability of the MIP/S-g-C_3_N_4_/SPR sensor (Appendix A).

## 4. Conclusions

In this work, a novel molecularly imprinted SPR sensor for FEN detection was presented and applied to rice samples. Firstly, the fabrication of sulfur-doped graphitic carbon nitride with zero waste was completed by a simple thermal poly-condensation method. Then, an FEN-imprinted SPR chip surface was developed using UV polymerization. To demonstrate the high imprinting selectivity, FEN non-dominated SPR chips were produced without the target molecule. According to the selectivity results, FEN analysis was carried out successfully with high selectivity and recovery. In addition, the values of LOQ of 1.0 ng L^−1^ and LOD of 0.30 ng L^−1^ showed a sensor design with high sensitivity. In conclusion, the prepared SPR sensor’s high selectivity, repeatability, reproducibility, and stability will provide the early diagnosis of diseases caused by pesticides and safe food consumption.

## Figures and Tables

**Figure 1 micromachines-15-00006-f001:**
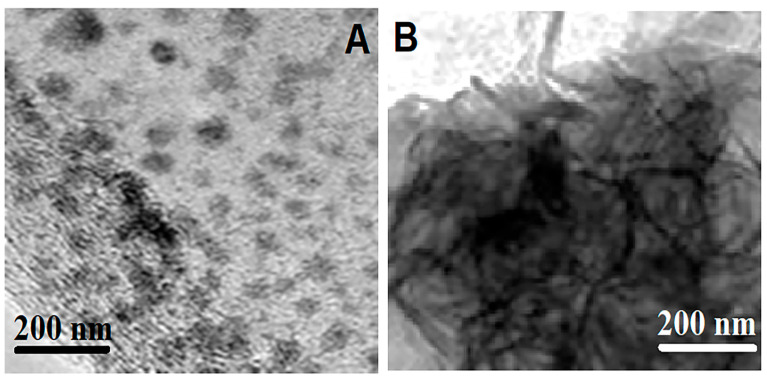
TEM images of (**A**) S-g-C_3_N_4_ and (**B**) g-C_3_N_4_.

**Figure 2 micromachines-15-00006-f002:**
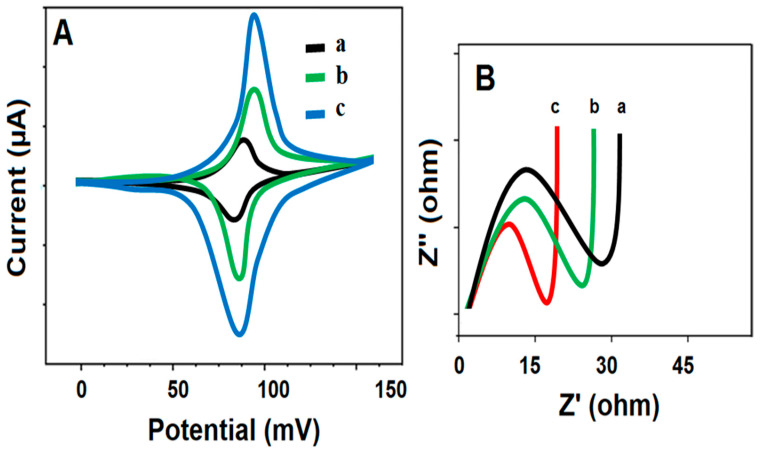
(**A**) CV curves and (**B**) EIS responses at (a) bare GCE, (b) g-C_3_N_4_/GCE, (c) S-g-C_3_N_4_/GCE (Redox probe: 5.0 mM [Fe(CN)_6_]^3−/4−^ containing 0.1 M KCl, potential scan rate: 50 mV s^−1^).

**Figure 3 micromachines-15-00006-f003:**
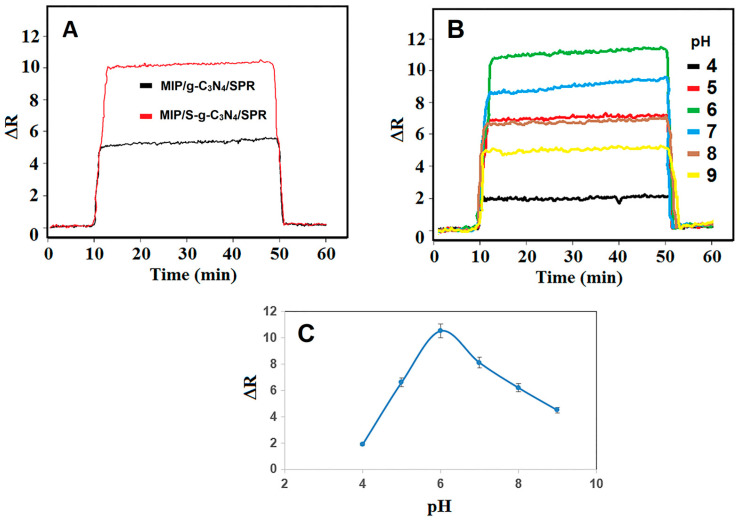
(**A**) SPR sensorgrams belonging to MIP/g-C_3_N_4_/SPR and MIP/S-g-C_3_N_4_/SPR chips for 10.0 ng L^−1^ FEN; (**B**) SPR sensorgrams belonging to different pHs of PBS for 10.0 ng L^−1^ FEN and (**C**) Effect of pH on SPR signals.

**Figure 4 micromachines-15-00006-f004:**
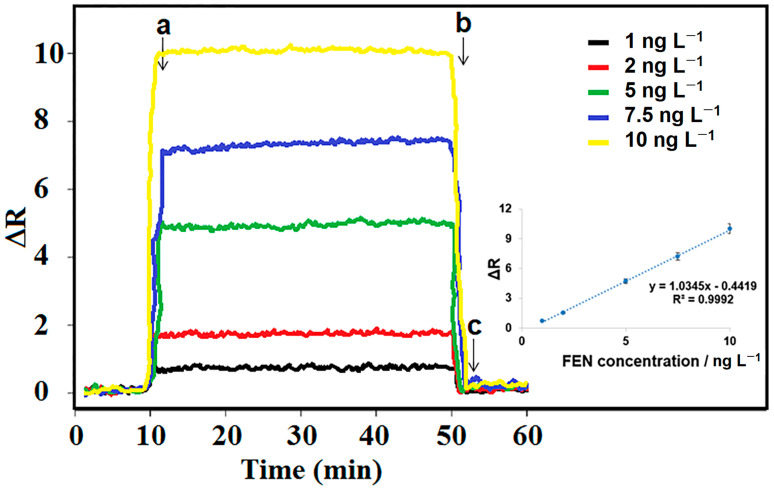
SPR sensorgrams, including the increasing concentrations of FEN standard solutions (from 1.0 ng L^−1^ to 10.0 ng L^−1^ FEN). Inset: Calibration curve at MIP/S-g-C_3_N_4_/SPR in presence of pH 6.0 of PBS: (a) adsorption; (b) desorption; (c) regeneration.

**Figure 5 micromachines-15-00006-f005:**
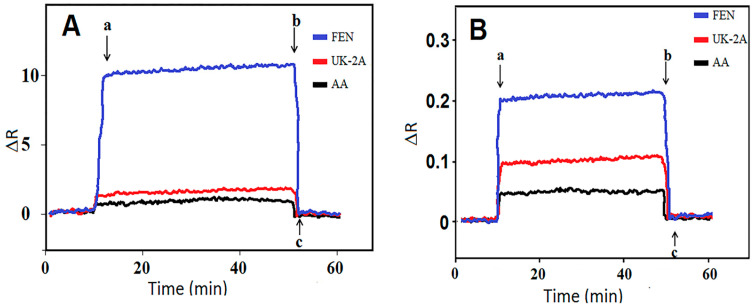
Selectivity tests: SPR sensorgrams of (**A**) MIP/S-g-C_3_N_4_/SPR and (**B**) NIP/S-g-C_3_N_4_/SPR in 10.0 ng L^−1^ FEN, 1000.0 ng L^−1^ UK-2A, and 1000.0 ng L^−1^ AA including pH 6.0 PBS. (a) adsorption; (b) desorption; (c) regeneration.

## Data Availability

Data are contained within the article.

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
