# Peer review of "Fenpicoxamid-Imprinted Surface Plasmon Resonance (SPR) Sensor Based on Sulfur-Doped Graphitic Carbon Nitride and Its Application to Rice Samples"

_micromachines, 2023, doi:10.3390/mi15010006_

Round 1

Reviewer 1 Report

Comments and Suggestions for Authors

The authors proposed an interesting work in SPR sensor using Sulfur-doped Graphitic Carbon Nitride and testing its application on rice samples. The stability, sensitivity and reproducibility seem good with this work. I recommend for publication after addressing following:

Introduction – 3rd paragraph: explanation for SPR is not written well. Please re-write with a clear description of optical mechanism. You can check and cite such references - https://doi.org/10.3390/ma15030792, https://doi.org/10.1021/acs.chemrev.7b00252, https://doi.org/10.1016/j.biosx.2022.100175

I suggest adding an inset graph (point graph) for fig. 1b to show the clear picture for red shift. It is not clear from the image how far the resonance peak has red-shifted!

Did authors measure any scattering data (LSPR)?

I suggest authors add a discussion part before conclusion (100 – 200 words) about how this work fares against the current and past literatures and what advantages can been seen from this.

Comments on the Quality of English Language

Please check basic grammar issues in few areas of the manuscript

Reviewer 2 Report

Comments and Suggestions for Authors

Summary:

This manuscript reports the development and utilization of a novel surface plasmon resonance (SPR) sensor. This newly designed sensor incorporates sulfur-doped graphitic carbon nitride (S- 17 g-C3N4) as the molecular imprinting material. The primary testing objective in this project is to use this sensor for quantitative analysis of Fenpicoxamid (FEN), a kind of propesticides in rice samples. The results demonstrate that the proposed sensor with high selectivity, repeatability, reproducibility and stability. The manuscript is well written but can be further polished.

Review:

The manuscript can be further improved, for example,

1) Can the authors describe a little bit more in detail for the advantages and disadvantages between the S-g-C3N4 and g-C3N4 both in theory and in experiments, including the cost and fabrication process?

Comments on the Quality of English Language

1) Page 5:

Table 3.

=> Where are Table 1-3? Do you mean Figure 3 instead?

2) Table S1

Are all of the samples are Rice? If so, then it doesn’t need to be listed as one column in the table, right?

3) Page 10:

In addition, the prepared SPR sensor based on MIP can be used be used more selectively and effectively in the presence of other agents with possible matrix effects.

=> “can be used be used” should be “can be used”

Reviewer 3 Report

Comments and Suggestions for Authors

Micromachines-2766742

Fenpicoxamid imprinted surface plasmon resonance (SPR) sensor based on sulfur-doped graphitic carbon nitride and its application to rice samples.

The manuscript discusses a novel application where the authors combined surface plasmon resonance (SPR) with a molecular imprinting material. They employed this system to detect a pesticide and the results were excellent. The article is well-written, and I believe it will be of great interest to the scientific community. However, before publication, the authors should make some corrections to improve the quality of the article.

Some notes:

In general, the quality of all the figures is low, and they appear to be blurred. The author did an excellent job of characterizing the surface using various techniques. However, the primary focus of the paper is on the detection of Fenpicoxamid using SPR and MIP. To improve the clarity of the manuscript, I suggest moving some of the characterization figures, such as XRD, FTIR, and electrochemical, to the supplementary information. Instead, figures S2 and S3 should be included in the main text.

Line 29: Abstract. The acronym for detection limit is DL, and LOD is the limit of detection. Both are used.

Line 203 and 211: The authors discuss Table 3, which is not included in the text.

I suggest the authors include the XPS survey spectra in the supplementary information file.

I suggest that the authors provide a more comprehensive explanation for the purpose of using electrochemical characterization.

I recommend that the author add a comparison sensorgram that shows signals of the same concentration. Furthermore, please show that the signal can be replicated using at least two sensors.

Based on my understanding, it would be beneficial if the conclusions are presented in a more detailed and comprehensive manner.

Round 2

Reviewer 1 Report

Comments and Suggestions for Authors

The current version of the manuscript is suitable for publication.